

# On the localization in strongly coupled ensemble data assimilation using a two-scale Lorenz model

Zheqi Shen[1], Youmin Tang[1,2], Xiaojing Li[1], Yanqiu Gao[1], and Junde Li[1]

[1]State Key Laboratory of Satellite Ocean Environment Dynamics, Second Institute of Oceanography, State Oceanic Administration, Hangzhou 310012, China
[2]Environmental Science and Engineering, University of Northern British Columbia, Prince George V2N 4Z9, Canada

**Correspondence:** Youmin Tang (ytang@unbc.ca)

**Abstract.** In the data assimilation of coupled models, the strongly coupled data assimilation (SCDA) is much more complicated than the weakly coupled data assimilation (WCDA), since it involves the cross-domain error covariances which could be very inaccurate when the ensemble size is small. In this study, the SCDA experiments are conducted using a two-scale Lorenz '96 model, which is a coupled system composed by two Lorenz '96 models in two domains have different temporal and spatial scales. A localization strategy is specially designed for the cross-domain error covariances when the ensemble adjustment Kalman filter (EAKF) is used for the coupled data assimilation (CDA) experiments. The formulas for computing the localization factors that can deal with multiple spatial scales and provide essential information are developed to improve the quality of analyses. The result shows that the SCDA can provides much more accurate estimation of the states than the WCDA when the localization for the cross-domain error covariances is used. Moreover, it is found that the advantage of the SCDA over the WCDA for this model is attributed to the assimilation of small scale observations into the coupled system, whereas the contribution of the assimilation of the large-scale observations to the coupled system is not obvious. This current study provides a possible strategy or idea for developing operational CDA using realistic coupled models.

## 1 Introduction

Data assimilation incorporates observations into numerical models to generate good estimates of the model states and accurate initial conditions for weather and climate predictions. However, sophisticated coupled models generally consist of several components that interact with each other through specific coupling mechanisms. Different model components may have different temporal and spatial scales, making CDA very difficult.

According to the degree of information exchanged, there are two levels of CDA, i.e., WCDA and SCDA. In WCDA, observations are incorporated into the same component without the exchange of information between components in the analysis step (Zhang et al., 2007). Thus, the coupling is completed by only dynamical models in the forecast stage. WCDA is an existing method that is increasingly being implemented in weather predictions and holds promise for improving sub-seasonal to





seasonal prediction systems (Lahoz and Schneider, 2014). However, the observed information cannot directly transfer across component boundaries in accordance with multiple dynamics and scales in WCDA, additional treatments such as additional analysis iterations are often required in order to increase the strength of the coupling in the analysis stage (Smith et al., 2018; Laloyaux et al., 2016).

SCDA provides a potential solution to the weaknesses of WCDA since it allows observations within one component of the system to directly affect the state estimates in other components. In SCDA, the coupling is done not only dynamically in the forecast stages but also statistically in the analysis stages. However, the SCDA method is still in its early stage and needs to be intensively studied further before its operational application. The WMO whitepaper (Penny et al., 2017) has identified the challenges for CDA. It strongly indicated that information propagation across model components with different spatiotemporal

scales is extremely complicated, and must be improved.

    In this study, we attempt to make some efforts towards meeting this goal using a two-scale Lorenz'96 (tsL96 hereafter) model (Lorenz, 1996). In the data assimilation community, the Lorenz '96 model is a simple model that is frequently used as a testbed for data assimilation methods (Anderson, 2003). This model represents an atmospheric variable at several equally spaced points around a circle of constant latitude. The tsL96 model has been constructed by coupling two systems, each of

which, aside from the coupling, obeys a suitably scaled version of the Lorenz '96 model. The model simulates the interaction between multiple temporal and spatial scales and is a perfect model to test and develop the SCDA methods (Luo and Hoteit, 2014).

    Many studies have been conducted with SCDA using simple models. For example, Zhang et al. (2012) used a simple coupled model consisting of a 3-variable Lorenz model and a slowly varying slab ocean to develop a SCDA scheme for enhanced

parameter correction. Han and Wu (2013) used an advanced version of this simple coupled model consisting of a Lorenz atmosphere and a pycnocline ocean model, which characterizes the interaction of media at different time scales in the climate system, to study the impact of the accuracy of cross-domain error covariance on the quality of SCDA. The cross-domain error covariance (also referred as coupling error covariance in literatures) measures the covarying strength of two variables residing in different media that have different scales. It plays a critical role in SCDA with ensemble-based data assimilation

methods (such as the Ensemble Kalman filter). Han and Wu (2013) concluded that the improvement from direct observational adjustments from a different model domain strongly depends on the accuracy of the evaluated cross-domain error covariance. The SCDA can be more effective than WCDA only if the ensemble size is large enough to evaluate the cross-domain error covariance correctly. That would be a very large number with respect to the model dimensions. However, their model uses only six model variables and does not simulate the spatial locations, which prohibits the use of localization method to improve the

accuracy of an evaluated error covariance.

    Luo and Hoteit (2014) have developed an ensemble Kalman filter with a divided state-space strategy for CDA using the tsL96 model system, which consists of two subsystems each controlled by a Lorenz '96 model with coupling terms. They conducted data assimilation experiments with respect to each subsystem, involving quantities from the subsystem itself and correlated quantities from other coupled subsystems. Using the ensemble transform Kalman filter (Bishop et al., 2001), the cross-domain

error covariance is used to generate the off-diagonal blocks of the Kalman gain for cross updates. They compared the WCDA



and SCDA frameworks, and made similar conclusions to Han and Wu (2013). However, they only considered the different time scales while ignoring the different spatial scales in their tsL96 model settings, and they used the same localization strategy for both in-domain and cross-domain error covariance, which is a lack of generality.

In this work, we conduct the SCDA experiment using the tsL96 model with different temporal and spatial scales. The
ensemble adjustment Kalman filter (EAKF) proposed by Anderson (2003) is used to assimilate the observations of both scales. In order to localized the cross-domain error covariance, we introduce new formulas to compute the localization factors which are used when assimilating the quantities from other coupled subsystems. The new localization strategy can better evaluate the cross-domain error covariance with limited ensemble size, that makes SCDA provide more accurate analyses than WCDA.

The paper is organized as follows. In section 2, the experiment settings are introduced with details of the tsL96 model and the
design of twin experiments. The EAKF method is also presented in section 2, while the details of corresponding localization strategy are discussed in section 3. Section 4 investigates the performance of the CDA experiments, showing the importance of localization in SCDA, and further promoting the SCDA method with flexible localization options for cross updates. Section 5 concludes the work and discusses some potential future studies.

## 2 Experiment Settings

### 2.1 Two-scale Lorenz'96 model

The two-scale Lorenz '96 model was originally introduced to simulate mid-latitude weather and to study the influence of multiple spatiotemporal scales on the predictability of atmospheric flows (Lorenz, 1996; Fatkullin and Vanden-Eijnden, 2004). It consists of $K$ slow-varying variables $\{X_k\}_{k=1}^{K}$ coupled to $J*K$ fast-varying variables $\{Z_{j,k}\}_{(j,k)=(1,1)}^{(J,K)}$ whose evolutions are governed by

$$\frac{dX_k}{dt} = X_{k-1}(X_{k+1} - X_{k-2}) - X_k + F - \frac{hc}{b}\sum_{j=1}^{J} Z_{j,k}, \tag{1}$$

$$\frac{dZ_{j,k}}{dt} = cbZ_{j+1,k}(Z_{j-1,k} - Z_{j+2,k}) - cZ_{j,k} + \frac{hc}{b}X_k, \tag{2}$$

where both $X_k$ and the $Z_{j,k}$ are assumed to be periodic, i.e., $X_{k+K} = X_k$ and $Z_{j,k+K} = Z_{j,k}, Z_{j+J,k} = Z_{j,k+1}$. The tsL96 model is an extension of the original Lorenz'96 model proposed by Lorenz (1996), which represents an atmospheric variable at $K$ equally spaced points around a circle with a constant latitude. The tsL96 model has been constructed by coupling two
subsystems, each of which, aside from the coupling, obeys a suitably scaled version of the Lorenz '96 model. The variables $X_k$ and $Z_{j,k}$ represent some atmospheric quantities discretized respectively into $K$ and $K*J$ sectors along the latitude circle. Figure 1 shows an example in which $K = 8$ and $J = 20$. It can be seen that the $X_k$ variables have a large spatial scale while the $Z_{j,k}$ variables have a small spatial scale. In addition, $Z_{j,k}$ for $j = 1, 2, \ldots, J$ are in the sub-domain of the domain that correspond to $X_k$. For convenience, we call Eq. (1) the large-scale model (L-model) with model variable $X$ and call Eq. (2)
the small-scale model (S-model) with model variable $Z$. The L-model and S-model variables are driven by quadratic nonlinear



interaction modeling advection, constant forcing, linear damping, and coupling between both models in the corresponding sectors. The constant parameters $b$, $c$ and $h$ represent the spatial and temporal scale ratios and coupling coefficient, respectively.

In this study, we set $K = 36$ and $J = 10$, so that each $X_k$ represents the average value of an atmospheric quantity over ten degrees of longitude, whereas each $Z_{j,k}$ represents the quantity of one degree of longitude. We also set $c = b = 10$ as in Lorenz (1996), implying that the S-model variables $Z_{j,k}$ tend to fluctuate ten times as rapidly as the L-model variables $X_k$, while their typical amplitudes are $1/10$ the size of the typical $X_k$ amplitudes. We let the coupling coefficient $h$ equal 1.0. We advance the model in time steps of 0.005 time units (TUs), which is equal to 36 minutes in reality. The constant forcing term is set to $F = 10$ to make both models vary chaotically. The integration starts from random initial values and runs 144000 steps ( 10 years) to allow fluctuations in the system to develop sufficiently. We use the integration results as the initial conditions for the data assimilation experiment. Figures 2a and 2c show the time series of the first variable of the L-model and S-model, respectively, while Figures 2b and 2d show a snapshot of all L-model and S-model variables at the 1000th model step. It can be clearly seen that these two models have different temporal and spatial scales.

## 2.2 Twin experiment setup

To evaluate the performance of CDA, a twin experiment is designed as follows. First, the model is integrated for 16000 model steps (or 400 days) using a step size of 0.005 TUs, starting from the initial conditions given in the previous section. Reference solutions are considered as true states for comparison.

The observations are generated by adding random noise to the true states. Figure 2a and 2b show that the S-model tends to fluctuate ten times as rapidly as the L-model, while figure 2c and 2d show that the amplitude of L-model variables is about ten times as much as the amplitude of S-model variables. Accordingly, the observational frequencies and the amplitudes of the errors are designed to be different for both models. The long term standard deviations (LT-STDs) of the variables of each model are also shown in figure 2c and 2d, respectively. The observational errors of each variable in each model are simulated by Gaussian random noise with STDs proportional to the LT-STDs. We assume that the STDs of the L(S)-observation errors are 30% of LT-STDs of L(S)-model. For the S-model, observations take place every 5 steps (3 hours). One of every two variables is observed. Specifically, we can only observe $Z_{j,k}$ if $j$ is an odd number. For the L-model, observations take place every 40 steps (1 day) for all the variables. Since each observation is assumed to be independent, the observational error covariance matrix $R$ is a diagonal matrix.

Similar to Luo and Hoteit (2014), the initial ensemble is generated by perturbing the initial conditions with random Gaussian noise of variance 1. The time series with data assimilation is evaluated against the true states at all steps (include both forecasts and analyses) for each model separately.

## 2.3 Ensemble adjustment Kalman filter

In this study, we employ the sequential version of the EAKF (Anderson, 2003) to perform the CDA experiment. We assume that the observational errors are independent here, which allows the EAKF to assimilate observations sequentially.



For each single scalar observation, the EAKF first computes increments in the observation space using the ensemble mean and variance and then computes increments for each state vector independently by regressing the observation space increments onto the state vector space. The state increments are used to update the prior ensemble, in which the localization technique can be employed to suppress the spurious long-distance correlations. We provide the details of localization in section 3 and show

the algorithm of EAKF as follows.

If $x$ denotes the state vector, $y^o$ denotes the single scalar observation with variance $\sigma_o^2$, and $h$ is the measurement operator; the EAKF algorithm first applies the operator $h$ to each ensemble sample of the state, producing the ensemble of prior estimates for the observation, namely,

$$y_{p,n} = h(x_{p,n}), n = 1 \ldots, N \tag{3}$$

Here, subscript $p$ indicates prior, and $n$ is the index of the ensemble member. The sample mean $\bar{y}_p$ and variance $\sigma_p^2$ of the prior estimate of the observation are computed.

Given the scalar observation value $y^o$ and the observational error variance $\sigma_o^2$, the product of the prior and the likelihood yields an updated estimate with variance

$$\sigma_u^2 = [(\sigma_p^2)^{-1} + (\sigma_o^2)^{-1}]^{-1} \tag{4}$$

and mean

$$\bar{y}_u = \sigma_u^2 \left( \frac{\bar{y}_p}{\sigma_p^2} + \frac{y^o}{\sigma_o^2} \right) \tag{5}$$

The updated ensemble estimate for $y$ is given by

$$y_{u,n} = \left( \frac{\sigma_u}{\sigma_p} \right)(y_{p,n} - \bar{y}_p) + \bar{y}_u, \quad n = 1, \ldots, N \tag{6}$$

which is computed by shifting the mean and linearly contracting the members to make the sample variance exactly $\sigma_u^2$. The

ensemble of observation space increments is defined as $\Delta y_n = y_{u,n} - y_{p,n}$.

The increments for each state vector entry are then computed independently by regressing the observation space increments onto the state vector using the prior joint ensemble sample statistics so that

$$\Delta x_{m,n} = \frac{\sigma_{x_m,y}}{\sigma_p^2} \Delta y_n, \quad n = 1, \ldots, N \tag{7}$$

where $\Delta x_{m,n}$ is the increment for ensemble member $n$ of state vector entry $m$, while $\sigma_{x_m,y}$ is the prior sample covariance of

state vector entry $x_m$ and $y$. The term $\sigma_{x_m,y}/\sigma_p^2$ is the form that the Kalman gain takes in the EAKF. The full Kalman gain is not required to be computed and saved in memory, so the computational requirement for the EAKF is affordable for large, complicated models.

The increment for each state vector is added to each ensemble member to update the prior ensemble. However, the localization technique that uses a distance-dependent factor $\rho$ can be used to suppress spurious long-distance correlations resulting from





an insufficient ensemble size. Using localization means that $x_{m,n}$ is updated by adding $\rho * \Delta x_{m,n}$. The details of determining the localization factor $\rho$ will be given in section 3.

The procedure is repeatedly applied for each scalar observation until all available observations within the data assimilation window are assimilated. That completes the analysis stage of an assimilation cycle.

In EAKF, the observations of each model are assimilated into the model system sequentially and state vectors are serially updated by each entry $x_{m,n}$, which means that the background error covariance matrix is not generated explicitly. Thus the numerical instability due to large condition numbers would not occur even though the model variables are with very different scales.

## 3   The localization strategy for CDA

### 3.1   Localization in EAKF

In ensemble-based data assimilation schemes, the ensemble size is often small due to limited computational resources. An insufficient ensemble size can result in spurious correlations between distant locations in the background covariance matrix and, thus, in the Kalman gain. Unless they are suppressed, these spurious correlations will cause observations at one location to affect the analysis of other locations of an arbitrary large distance away, in an essentially random manner. This needs to be

remedied by the localization method.

Localization is introduced to eliminate the background error covariance associated with remote observations. There are two localization approaches that are frequently used in ensemble data assimilation (Farchi and Bocquet, 2018). In the first approach, independent analyses are performed for each grid point by using only the observations sites that influence this point. This approach is known as domain localization. The second approach is used when an analysis is performed for each

observation site. When assimilating an observation of a site, only grid points within a certain distance are updated while distant grid points remain unchanged. This approach is also referred as observation localization.

The EAKF uses observation localization, in which a continuous function whose values are inversely proportional to the distances from observation sites is employed to generate the multiplication factor for the state increments. This Gaspri-Cohn function (Gaspari and Cohn, 1999) is the most widely used function to cut off long-distance correlations, i.e.,

$$\rho = \Omega(d,c) = \begin{cases} -\frac{1}{4}(\frac{d}{c})^5 + \frac{1}{2}(\frac{d}{c})^4 + \frac{5}{8}(\frac{d}{c})^3 - \frac{5}{3}(\frac{d}{c})^2 + 1, & 0 \leq d \leq 2c; \\ \frac{1}{12}(\frac{d}{c})^5 - \frac{1}{2}(\frac{d}{c})^4 + \frac{5}{8}(\frac{d}{c})^3 + \frac{5}{3}(\frac{d}{c})^2 - 5(\frac{d}{c}) - \frac{2}{3}(\frac{d}{c})^{-1}, & c \leq d \leq 2c; \\ 0, & d \geq 2c \end{cases} \tag{8}$$

where $d$ represents the distance between the observation site corresponding to $y^o$ and model grid corresponding to the entry $x_m$. The parameter $c$ relates to the decorrelation length. In actuality, the localization factor $\rho$ equals zero when $d$ is larger than $2c$, whereas $\rho = 1$ for $d = 0$. In addition, the factor decreases monotonically with the distance $d$. Using the localization approach, the EAKF updates the prior state estimates by

$x_{m,n} = x_{m,n}^{(p)} + \rho * \Delta x_{m,n}, n = 1, \ldots, N$ (9)





in which $x_{m,n}^{(p)}$ indicates the m-th entry of the prior ensemble member.

In this work, the EAKF method is performed on the tsL96 model. Typically, we can use $|k_1 - k_2|$ to represent the distance between $X_{k_1}$ and $X_{k_2}$ in the L-model. Due to the periodicity of $X_k$, if $|k_1 - k_2| > K/2$, it should be replaced by $K - |k_1 - k_2|$. Similarly, we can also define the distance between $Z_{j_1,k}$ and $Z_{j_2,k}$ as $|j_1 - j_2|$ in the S-model, while making use of the periodic

conditions $Z_{j,k+K} = Z_{j,k}$ and $Z_{j+J,k} = Z_{j,k+1}$. However, both models have different spatial resolutions, as Figure 1 shows; thus, the unit distance definitions are different for both models. For example, the distance value of 10 implies 100 degrees of longitude in the L-model, but 10 degrees of longitude in the S-model. As a consequence, the localization parameters and the corresponding localization factor for both models should be different.

Keep in mind that we are using the observations $[Y_X; Y_Z]$ to update the model variables $[X; Z]$, in which $Y_X$ and $Y_Z$ are observations of the L-model and S-model respectively (call L-observation and S-observation hereafter). As eq. (7) indicates, the correlation covariance between model grids and observation sites are required to compute the increment amount. When multiple models are involved, the correlation covariance of $[Y_X; Y_Z]$ and $[X; Z]$ can be represented by the block matrix below

$$\Sigma = \begin{bmatrix} \Sigma_{xx} & \Sigma_{zx} \\ \Sigma_{xz} & \Sigma_{zz} \end{bmatrix}.$$

$\Sigma_{xx}$ and $\Sigma_{zz}$ are covariance matrices for the model grids and observation sites within the same L-model and S-model, respec-

tively. Whereas $\Sigma_{zx}$ and $\Sigma_{xz}$ are covariance matrices for model grids and observation sites in different models.

Combining eq. (7) and eq. (9), it is easy to find that the localization factor is applied on the covariance matrix $\Sigma$ by taking Shur product. Accordingly, The localization factors can be written in a same matrix form $P$, which is also divided into four blocks.

$$P = \begin{bmatrix} P_{xx} & P_{zx} \\ P_{xz} & P_{zz} \end{bmatrix}.$$

Among these blocks, $P_{xx}$ is a matrix of the localization factors for L-model, which is a $K * K$ diagonal-constant matrix with each value of localization factor $\rho_{xx}$ in each diagonal. $P_{zz}$ is a matrix of the localization factors for S-model, which is a $(KJ) * (\frac{KJ}{2})$ diagonal-constant matrix when only half of the S-variables are observed. The off-diagonal blocks $P_{xz}$ and $P_{zx}$ are respectively constituted with the localization factors $\rho_{xz}$ and $\rho_{zx}$ for cross-domain error covariances. And they should

have $(KJ) * K$ and $K * (\frac{KJ}{2})$ elements respectively. Particularly, $\rho_{xz}$ is used to localize the S-observation when updating L-variables, and $\rho_{zx}$ is used to localize the L-observation when updating S-variables. In sections 3.2 and 3.3, we are determining the $\rho_{xx}$ and $\rho_{zz}$ by the sensitive experiments, and computing $\rho_{xz}$ and $\rho_{zx}$ by some newly developed formulas.

### 3.2  In-domain localization

There has been several recent studies on optimal localization parameters, which can minimize the data assimilation errors due to

spurious long-distant correlation. For example, Menetrier et al. (2015) found the optimality criteria for linear and Schur filtering of covariance, which works well with the domain localization. Kirchgessner et al. (2014) studied the criteria for an optimal localization radius in Ensemble Kalman filters, and applied in the Lorenz'96 model. However, the two-scale Lorenz'96 model





has some different properties which relate to the coupling, and the problem related to optimal localization could beyond the scope of this work. To simplify the discussion, we use the twin-experiment with constant ensemble size and different choices of localization parameter to determine the parameter $c$ in eq. (8), which minimizes the root-mean-squared-errors averaged over the whole data assimilation period (call MRMSE hereafter).

To determine the in-domain localization parameter for the observation of each model, we use single model data assimilation experiments. To be specific, we use the L-model data assimilation experiment to determine the optimal $c$ value for $\rho_{xx}$ and use the S-model data assimilation experiment to determine the optimal $c$ value for $\rho_{zz}$. The single model data assimilation experiment is conducted by using the reference solutions of the other model as external terms and assimilating only the observations of the same model.

First, we perform the L-model data assimilation while using the true values of $\{Z_{j,k}\}_{j=1}^{J}$ in the external forcing. Only the L-observations are assimilated to update the L-variables, and the RMSE of all $X_k$ against their true states are computed at every model step. The MRMSE is calculated to measure the performance of EAKF with different localization parameters. In this experiment, different localization parameters are tested in a range of $\{1, 2, 4, 8, 16, 32\}$, and with different ensemble size of $N = 20, 40, 80, 160$ and 320. The assimilation duration is 16000 model steps, and observations are available every 40 model

steps, as mentioned in section 2.2. It is also noteworthy that the EAKF uses an inflation method, which increases the ensemble spread by multiplying it by a positive number slightly larger than one. For simplicity, we use the inflation method with a fixed factor $\alpha$, which is tuned by sensitivity experiments and valued at 1.01 eventually.

Figure 3a shows the MRMSE of the L-model assimilation using different localization parameters with different ensemble sizes. Localization is mainly used to suppress spurious long-distance correlations, which will introduce large analysis errors

and are very likely to occur when the ensemble size is small. The optimal value of the localization parameter is dependent on the ensemble size. It should be neither too large nor too small so that it can update as many variables as possible while not introducing spurious correlations. In Figure 3a, it can be seen that the localization parameter value of 32 corresponds to the smallest MRMSE among these choices of localization parameters. It is consistent with the result in Kirchgessner et al. (2014) for the 40-dimensional Lorenz '96 model. Since the L-model has only 36 variables, the case with $c = 32$ is very similar to the

no localization scenario. We would not consider the value of $c$ beyond 32 because it will gain very little error reduction. In order to simplify the discussion, we will use the parameter $c = 32$ for all ensemble sizes to perform the EAKF method for the following CDA experiment.

Secondly, we perform the S-model data assimilation while using the true values of the $X_k$ as an external forcing in the integration. The S-observations are available every 5 model steps, and only half of the $Z$ variables are observed. The MRMSE

of all $Z$ variables against true states are calculated and compared in Figure 3b. In Figure 3b, it seems that the optimal localization parameter is 8 for $N \geq 40$. Considering that there are 360 variables in the S-model, c=8 essentially restricts the influence of S-observations in a very limited range.

In Figure 3b, the optimal localization parameters for S-model data assimilation prefer the values smaller than 8 even though a large ensemble size (such as $N = 320$) is used. This is an interesting result that seems inconsistent with our previous knowledge

about localization. As we know, localization can be regarded as a remedy for the insufficient ensemble size in ensemble-based





methods. Theoretically, as the ensemble size increases, larger localization parameters can be used to allow each observation to update more model variables without introducing spurious correlations. To explain Figure 3b, we examine the properties of the coupling in tsL96 model. Lorenz (1996) revealed some typical behaviors of this model; namely, there are seven active areas for the large-scale $X_k$, generally 30 or 40 degrees wide, which fluctuate strongly in width and intensity. This can be seen in

Figure 4, which reproduces the same results in Lorenz (1996), showing the variation and interaction of $X$ and $Z$ variables over a whole day (40 model steps). It shows that the convective activity (which relates to the small-scale variables $Z_{j,k}$) is patently strongest in these active areas and rapidly dies out as it leaves an active area. It indicates that one small scale observation can theoretically influence up to 30 or 40 degrees of longitude in length and cannot influence the small-scale variables of other active areas. Coincidentally, $c = 8$ corresponds to 15 degrees of longitude in length from both sides. Thus, the optimal choice

of $c = 8$ for the S-model is due to the interaction between multiple scales in the tsL96 model system, and we can use $c = 8$ for the S-model for ensemble sizes larger than 20. However, when $N = 20$, $c = 4$ is chosen due to the requirement to suppress spurious correlations.

As a result, the localization parameter for computing $\rho_{xx}$ is $c = 32$, while the localization parameter for computing $\rho_{zz}$ is $c = 8$ or $c = 4$ (particularly for $N = 20$).

### 3.3 Cross-domain localization

The in-domain localization can be used to update the L-variables and S-variables separately in a WCDA framework, in that case, all $\rho_{xz} = \rho_{zx} = 0$. To perform the SCDA experiment, the cross-domain localization that addresses the cross update between two models of different spatial scales must be developed.

Because both models have different spatial resolutions, it is very difficult to compute the distance between a S-model grid

point and a L-model observation site, and vice versa. Thus, we cannot directly use the localization factors $\rho_{xx}$ to localize L-observation when updating S-variables; nor can we use $\rho_z z$ to localize S-observation when updating L-variables. New formulas are required to compute the $\rho_{xz}$ and $\rho_{xz}$ when the cross update is enabled.

In the tsL96 model, which is governed by Eq. (1) and Eq. (2), the interactions between the two models are implemented by the coupling terms, i.e., the last terms of the right hand sides of Eq. (1) and (2). As seen in the L-model equation, the

S-variables affect the L-variables by their average over a unit distance of the L-model. Here, the amount of a unit distance of the L-model is 10 degrees in longitude, which is equivalent to 10 unit distance of the S-model, as discussed above. Thus, the impact of the whole S-model on a specified L-variable (e.g., $X_{k-1}$) can be measured by the average over 10 corresponding S-variables (e.g., $\{Z_{j,k-1}\}_{j=1}^{10}$), as indicated by the large downward arrow in the schematic diagram in Figure 5a. Assume that an observation of $Z_{j,k}$ is assimilated into the S-model, $\rho_{zz}$ can be computed for each Z-variable to localize the observation.

When this observation is also used to update $X_{k-1}$, $\rho_{zx}$ need to be computed from $\rho_{zz}$ to localize the corresponding entries in the cross-domain error covariance. Considering that $X_{k-1}$ is directly connected to $\{Z_{j,k-1}\}_{j=1}^{10}$, a straightforward method to compute $\rho_{zx}$ for $X_{k-1}$ is to average the $\rho_{zz}$ for $\{Z_{j,k-1}\}_{j=1}^{10}$ over a distance unit of the L-model. It is equal to say that $\rho_{zx}$ for





$X_k$ can take the value of $1/J \sum_{j=1}^{J} \rho_{zz}$, in which all $\rho_{zz}$ for $\{Z_{j,k}\}_{j=1}^{J}$ are included. In matrix notations, we can write

$$P_{zx} = \frac{1}{J} \mathbf{ones}(1,J) \otimes I_K * P_{zz}, \tag{10}$$

where $\mathbf{ones}(1,J)$ is a row vector, $I_k$ is the identity matrix with rank $K$, and $\otimes$ represent the tensor product. Using Eq. (10), we compute the average of $P_{zz}$ entries over each $k$-th sector, and compress the spatial dimensions from $(KJ)*(\frac{KJ}{2})$ to $K*(\frac{KJ}{2})$.

On the other hand, in the S-model equation (2), the forcing item from one-unit distance of the L-model can drive all these S-variables spanned in the unit distance. That implies the $Z_{j,k}$ variables with the same $k$ are driven solely by $X_k$, and each L-variable can have equal effects on these $Z_{j,k}$ variables, as indicated in Figure 5b. When an observation of $X_k$ is used to update the S-model, the $Z_{j,k}$ variable with the same $k$ can use the same $\rho_{xz}$, because they are indistinguishable in the observation's scale. A rational choice of $\rho_{xz}$ for $Z_{j,k}$ could be the same $\rho_{xx}$ for $X_k$, since they are connected closely by the coupling

mechanism. In matrix notations,we can write

$$P_{xz} = P_{xx} \otimes \mathbf{ones}(J,1), \tag{11}$$

where $\mathbf{ones}(J,1)$ denotes a column vector. In Eq. (11), we use the tensor product to extend the spatial dimensions, making each L-variable has equal effect on the corresponding S-variables in the same sector.

## 4   CDA experiments

In this section, we perform the CDA experiments using the twin experiment setup and EAKF method with localization techniques described in the proceeding sections. The WCDA involves only the localization factors $\rho_{xx}$ and $\rho_{zz}$, which are determined by individual model data assimilation experiments as discussed in section 3.2. The localization factors $\rho_{xz}$ and $\rho_{zx}$ that are derived by the method presented in section 3.3 are additionally used for cross updates in the SCDA.

    The CDA experiment results are shown in Figure 6. Since both models are with different scales, the RMSE scaled by the climatology mean would better evaluate the performance of CDA methods on each model, which denotes

$$\frac{1}{K} \sum_{k=1}^{K} (\frac{X_k^{\mathbf{assim}} - X_k^{\mathbf{true}}}{X_k^{\mathbf{clim}}})^2$$

and

$$\frac{1}{K*J} \sum_{j=1}^{J} \sum_{k=1}^{K} (\frac{Z_{j,k}^{\mathbf{assim}} - Z_{j,k}^{\mathbf{true}}}{Z_{j,k}^{\mathbf{clim}}})^2$$

for L-model and S-model respectively. The mean of the scaled RMSE over the assimilation period (called MS-RMSE) is

computed against the ensemble size in Figure 6a and 6b. It can be seen that the SCDA produces much more accurate analyses than WCDA in both models. In particular, as the ensemble size increases, the advantage of SCDA becomes substantial.

    In addition, we also used a metric called the 'coefficient of efficiency' (CE) from Nash and Sutcliffe (1970) to measure the performance of SCDA and WCDA for the entire coupled system. The CE is defined below (Tardif et al., 2015)

$$CE = 1 - \frac{\sum_{t=1}^{T} (x_t^{\mathbf{true}} - x_t^{a})^2}{\sum_{t=1}^{T} (x_t^{\mathbf{true}} - (\overline{x^{\mathbf{true}}})^2)}$$



where $x_t^a$ could be either the L-model variable $X_k$ or the S-model variable $Z_{j,k}$, and $T = 16000$ is the data assimilation duration. It is obvious that perfect analyses have $CE = 1$, and a smaller $CE$ value implies fewer climatology information is contained. The mean value of CE is shown in Figure 6c, which confirms that SCDA analyses contain more information than those of WCDA.

For a further examination, in Figure 7 we show the data assimilation errors for L-variables and S-variables for WCDA and SCDA, respectively. The ensemble size is $N = 80$, with which SCDA can provide much more accurate estimates than WCDA, as indicated in Figure 6. It is clearly observable that SCDA has much smaller analysis errors than WCDA almost in every stage. Considering that the only difference between WCDA and SCDA algorithms is the cross-update from a different scale, the L-model must benefit from directly assimilating the S-observations very much. On the other hand, it is shown that the analysis errors for S-model are highly correlated to the state pattern, i.e., the errors are large when the variation of S-model is large, and vice versa. But there is no significant difference between the analysis errors of SCDA and WCDA for the S-model. It is possible that the smaller MS-RMSE with SCDA in figure 7b is due to the coupling from more accurate L-model variables.

To clarify this, we use different CDA frameworks for different observations in the following experiment. We consider four possible scenarios that differ from one another depending on the CDA levels (weakly or strongly) that are used for the observations, as listed in Table 1. Scheme 1 and scheme 4 are essentially equivalent to the WCDA and SCDA used in the proceeding sections, respectively. Scheme 2 represents the situation in which only L-observations can update both L-model and S-model, whereas scheme 3 corresponds to the case in which only S-observations can update the variables of both models.

Figure 8 compares the performances of these schemes. Again, we present the MS-RMSEs and mean CE as functions of ensemble size. In Figure 8a - c, we use the former settings of observation errors, i.e., the STD of the L(S)observation errors is $30\%$ of the mean L(S)-model LT-STD. In Experiment 2, whose results are shown in Figure 9d – f, the STD of the S-observation errors is doubled, i.e., the STD of the random S-observation errors is $60\%$ of the LT-STD while that of the L-observation remains $30\%$.

One interesting feature in Figure 8a-c is that the MS-RMSE and mean CE of CDA scheme 2 are very close to those of the WCDA. Meanwhile, the MS-RMSE and mean CE of CDA scheme 3 are almost identical to those of the SCDA. In Experiment 2, we have increased the uncertainty of the S-observations. Figure 8d-f show that even though S-observations are relatively much poorer than L-observations, this feature can also be found.

It evidently indicates that L-observations are not able to effectively update the S-variables directly, so assimilating the L-observations in the framework of SCDA barely improves the accuracies of the analyses. Even worse, when the ensemble size is small, strongly-coupled assimilated L-observations could introduce noise to the S-variables, making the S-model analyses worse than that of WCDA. That can be seen from Fig 8b when $N = 20$. As a result, we can conclude that the reduction of errors using SCDA is mainly due to the strongly CDA of the S-model observations. This result is easy to understand since the L-model observations do not contain the high-frequency information that corresponds to the S-model variables. Thus, a more logical and computational-economic CDA framework for the tsL96 model is CDA scheme 3, i.e., assimilating L-model observations with WCDA while assimilating S-model observations with SCDA. A more realistic application which assimilates the atmospheric observations into the ocean using SCDA method in Sluka et al. (2016) also supports this finding.





As previously mentioned, the performance of SCDA highly depends on the estimate accuracy of the cross-domain error covariance using ensemble members. If the cross-domain error covariance is accurately estimated, the observations of the other model component can provide essential information and improve the quality of analysis, as shown in Figures 6 and 7. Otherwise, the observations would mainly act as a source of noise and degrade the analyses. Only when a proper localization

strategy is applied, the spurious long-distance correlations could be suppressed, leading to an accurate estimation of the coupling error covariance with an affordable ensemble size. To show the importance of the localization strategy in the coupling error covariance, we compare the performance of SCDA scheme 3 with the case of 'no cross localization', in which we keep the same localization strategy for the in-domain update but no localization for cross-domain update (i.e., one S-observation can update all L-variables equally, or $P_{xz} = 0$, $P_{zx} = \mathbf{ones}(K, \frac{KJ}{2})$). It is equivalent to the case that all $\rho_{xz} = 0$ and $\rho_{zx} = 1$.

The MS-RMSE and mean CE are computed and compared in Figure 9. As shown in this figure, when the ensemble size is smaller than 320, the no cross localization method performs so poorly that it cannot provide any useful information. This indicates that the no cross localization method fails completely when the ensemble size is small. In this situation, the cross-domain error covariance is dominated by noise; thus, poor observation increments are produced. However, when $N \leq 320$, the ensemble size is large enough to estimate the coupling error covariance correctly, such that the no cross localization method

can be comparable to SCDA scheme 3.

## 5   Conclusions

The interaction between multiple spatial and temporal scales usually makes the CDA very challenging. In recent years, research on CDA methods has become a very hot topic and has attracted broad attention. While the WCDA is currently a main algorithm used in the operational CDA system, the SCDA is becoming intensively studied due to its inherent advantage over the WCDA.

A crucial issue in the SCDA is the construction of cross-domain error covariance, which is used in the cross update between different model components. The considerable studies have shown that the accuracy of the coupling error covariance is the key factor to improve the SCDA quality. For an ensemble-based assimilation algorithm, the estimate accuracy of the covariance is largely determined by the localization scheme that can well alleviate the impact of spurious observations inherent to limited ensemble size.

The challenge of constructing a localization scheme for the cross-domain error covariance in the SCDA is the presence different spatial and temporal scales in the model components. There has been little research about how the large-scale observation impact on small-scale model states and vice versa. In this study, we use the two-scale Lorenz '96 model, which acts as a test bed, to study these important issues.

The first issue is that the two subsystems in the tsL96 model have different scales, such that the localization factors, which

are used to suppress spurious long-distance correlations due to small ensemble sizes, should be different. In this work, we performed various sensitivity assimilation experiments for each subsystem to determine their individual localization parameters. The parameters that result in the smallest MRMSE are used to construct the localization factors for the update within the same subsystem.



The second issue is that the cross-update localization factor, which is exclusively for SCDA, is technically studied. A localization strategy was developed to compute the localization factors for cross updates between the large-scale and small-scale subsystems. The factors were formulated according to the coupling terms in the model equations and have much smaller analysis errors than WCDA with very small ensembles. Specifically, the localization factor for using a small-scale observation to

update a large-scale variable is to use an averaged localization factors of small-scale model itself over the distance equivalently to a unit distance of large-scale model; whereas the localization factors for using a large-scale observation to a update small-scale variable is simply taken from the localization factor of large-scale model itself. With this localization strategy, we have shown that SCDA can provide much more accurate analyses than WCDA.

In addition, we compared four scenarios that differentiated from each other depending on the CDA levels (weakly or

strongly) that were used for the observations. It was found that the advantage of SCDA over WCDA is mainly attributed to the strong assimilation of small scale observations, while the benefit from the strong assimilation of large scale observations is not obvious.

The localization techniques used in this work depend on the model equations to some extent, and the comparison results are derived based on the twin experiments with a simple coupled model. Realistic coupled models and observations are far more

complicated; therefore, the localization schemes of SCDA for these coupled models are still challenging. However, the Lorenz model has been used as a classic test bed to develop and propose new methods and algorithms in the field of data assimilation. The current study provides a possible strategy or idea for this challenge and guidance on a possible way to proceed. A further study is still required using realistic models, which we will pursue in the next step of our study.

*Acknowledgements.* This work is supported by grants from the National Key Research and Development Program (2017YFA0604202), Basic

Science Foundation of the Second Institute of Oceanography (JG1619), the National Program on Global Change and Air-Sea Interaction (GASI-IPOVAI-06), the National Natural Science Foundation of China (41321004, 41606012).



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





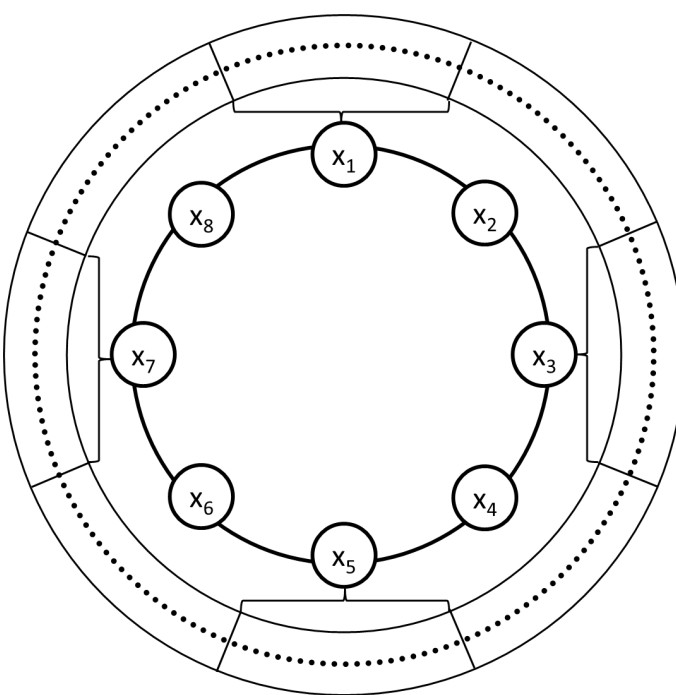

**Figure 1.** The physical meaning of the two-scale Lorenz '96 model, where the variables $X$ (circles) and $Z$ (dots) represent some atmospheric quantities discretized respectively into $K$ and $K*J$ sectors along the latitude circle. In this example, the values $K = 8$ and $J = 20$ are picked for a better illustration.

**Table 1.** Four CDA schemes based on using different levels of CDA for different observations, and one 'no cross localization' scenario for comparison.

|  | Level of CDA for L-obs. | Level of CDA for S-obs. | Localization matrix for cross update |
|---|---|---|---|
| CDA scheme 1 | Weak | Weak | $P_{xz} = 0,\ P_{zx} = 0$ |
| CDA scheme 2 | Strong | Weak | $P_{xz} = P_{xx} \otimes \mathbf{ones}(J,1),\ P_{zx} = 0$ |
| CDA scheme 3 | Weak | Strong | $P_{xz} = 0,\ P_{zx} = \frac{1}{J}\mathbf{ones}(1,J) \otimes I_K * P_{zz}$ |
| CDA scheme 4 | Strong | Strong | $P_{xz} = P_{xx} \otimes \mathbf{ones}(J,1),\ P_{zx} = \frac{1}{J}\mathbf{ones}(1,J) \otimes I_K * P_{zz}$ |
| 'no cross localization' | Weak | Strong | $P_{xz} = 0,\ P_{xz} = \mathbf{ones}(K, \frac{KJ}{2})$ |

Scheme 1 is equivalent to WCDA while scheme 4 is equvalent to SCDA mentioned in this paper.





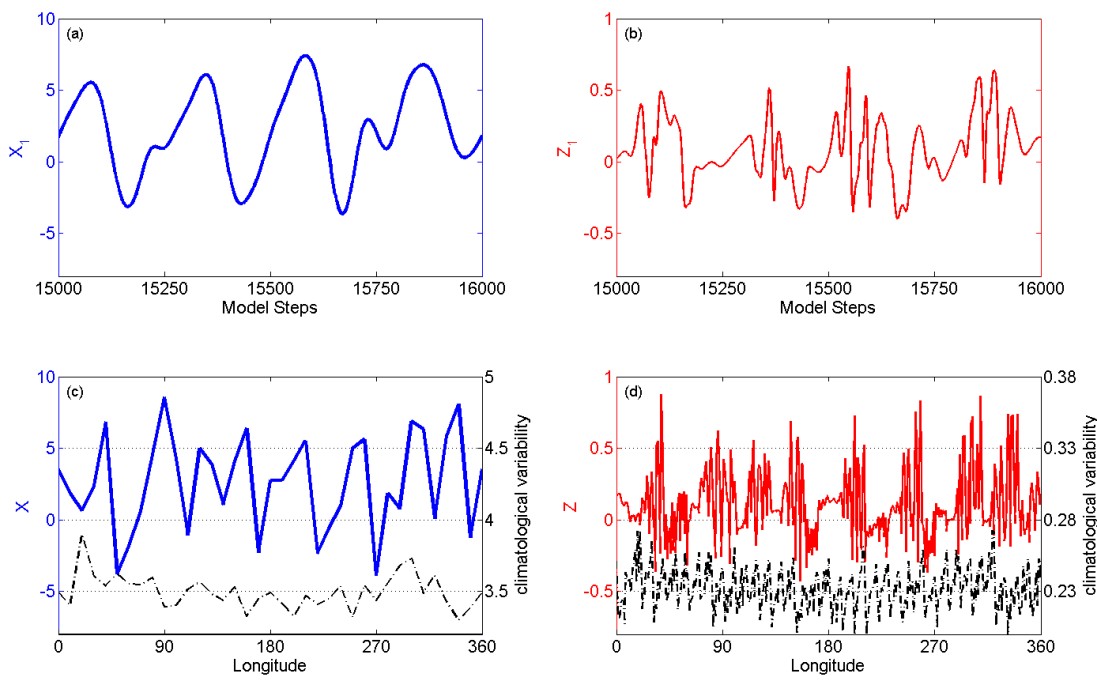

**Figure 2.** The time series for the variables $X_1$ (a) and $Z_{1,1}$ (b) for the last 1000 model steps, and the state vectors $X$ (c) and $Z$ (d) at the $16000^{\text{th}}$ model step. The black dashed line indicates the climatological variability of each model



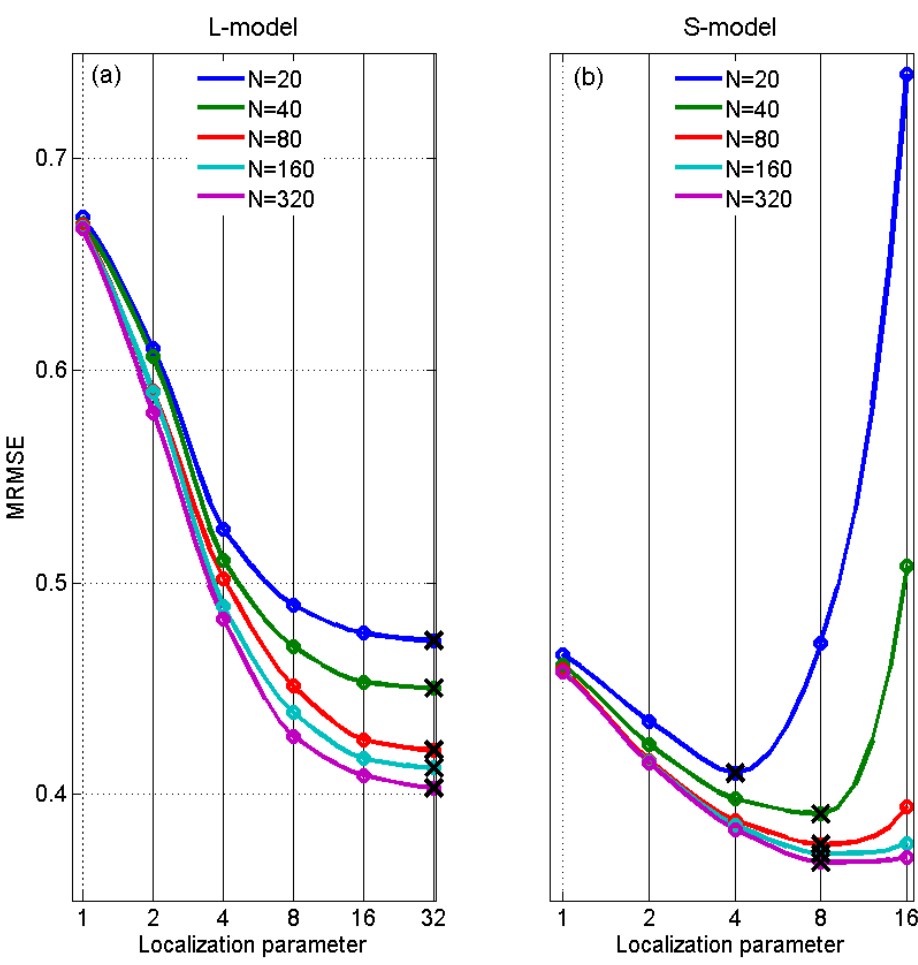

**Figure 3.** The average RMSE over 16000 model steps against localization parameters with different ensemble sizes in L-model data assimilation (a) and S-model data assimilation (b).



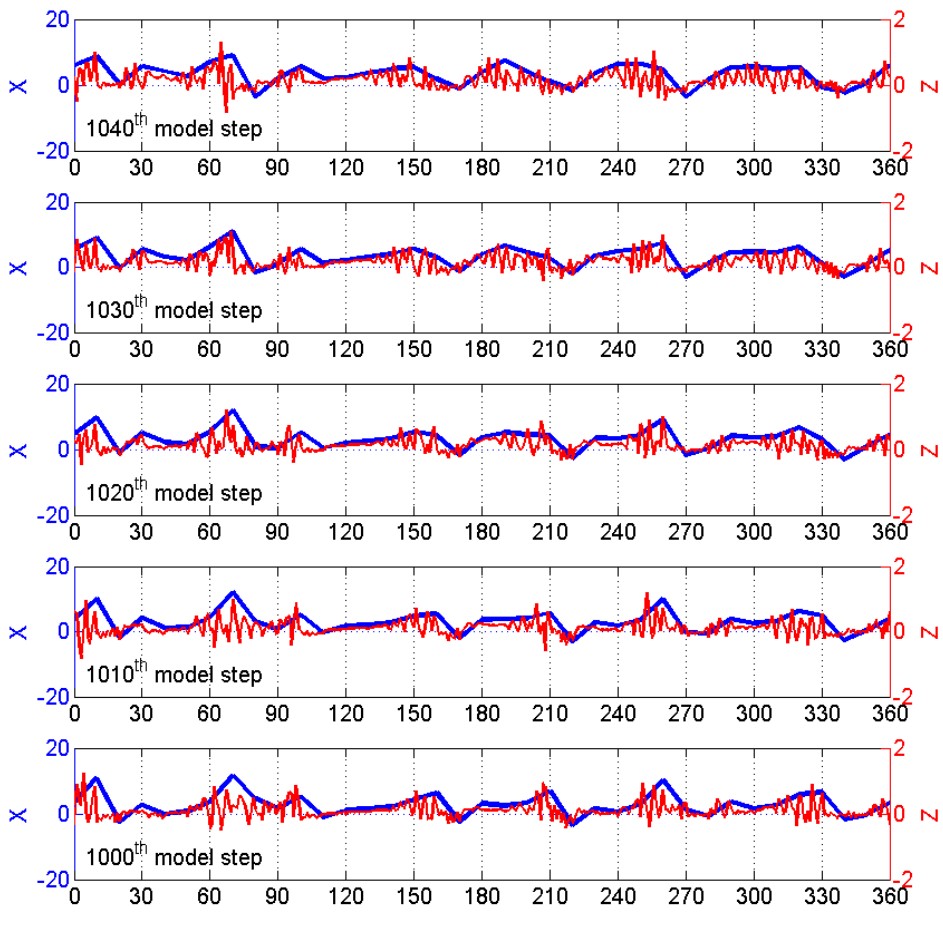

**Figure 4.** The evolution of L-model (blue) and S-model (red) variables over 24 hours (40 model steps).



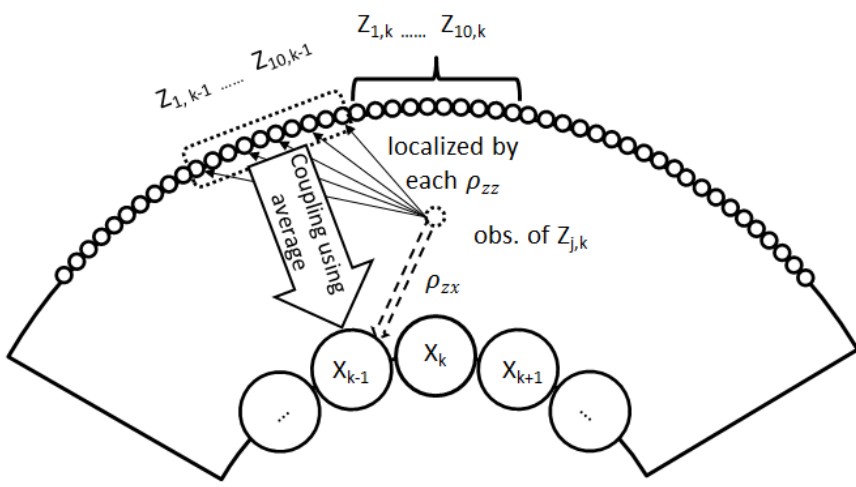

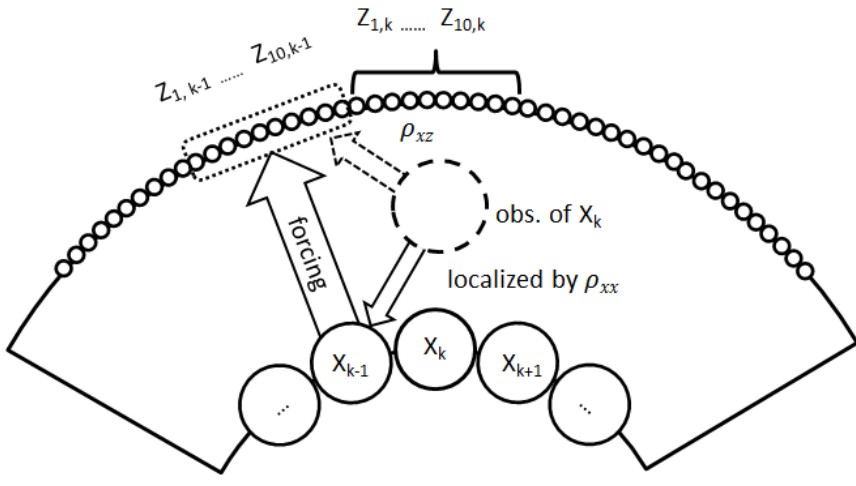

**Figure 5.** The schematic diagram for determining the localization factors for using S-observations to update L-variables (a) and using L-observations to update S-variables (b).

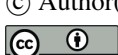
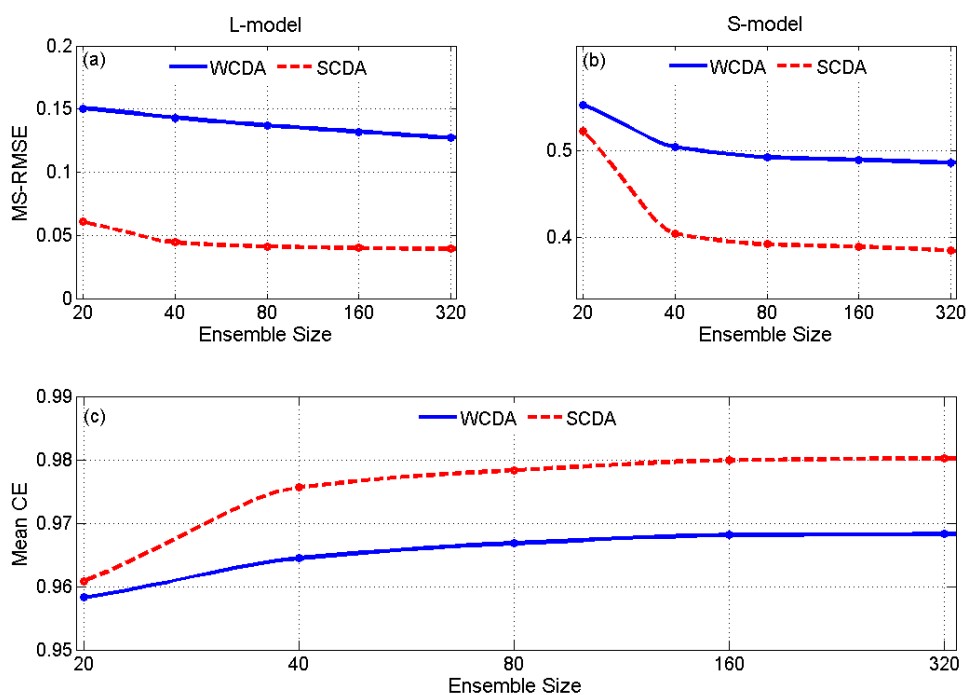

**Figure 6.** The MS-RMSE of L-model (a) and S-model (b) and the mean CE of the coupled system (c) against different ensemble sizes with WCDA and SCDA.


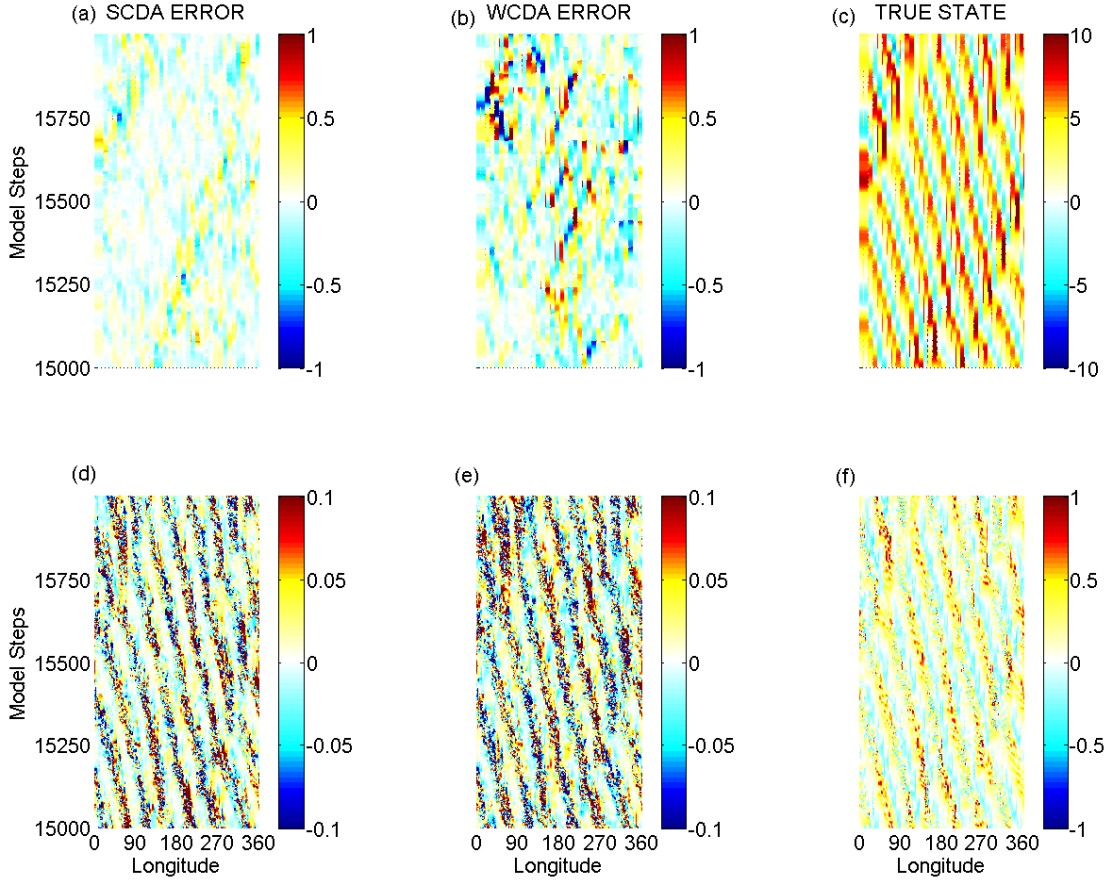

**Figure 7.** The analysis errors of L-model using SCDA (a) and WCDA (b), and the true states of L-model (c). The analysis errors of S-model using SCDA (d) and WCDA (e), and the true states of S-model (f)



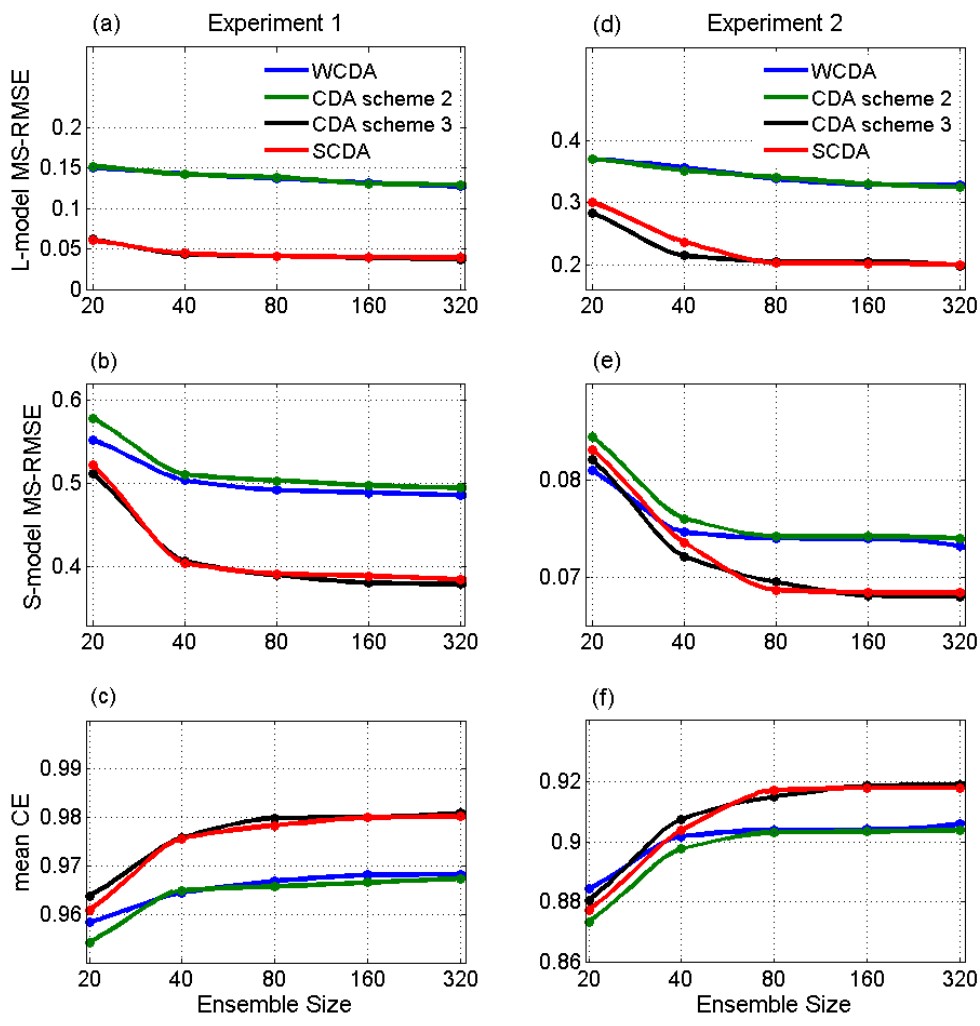

**Figure 8.** The MS-RMSE of L-model (a, d) and S-model (b, e) and the mean CE of the coupled system (c, f) against different ensemble sizes when the different CDA schemes in Table 1 are used. Experiment 1 and experiment 2 use different S-observation errors.

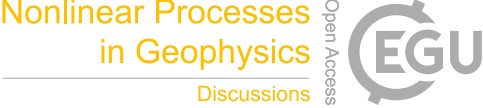



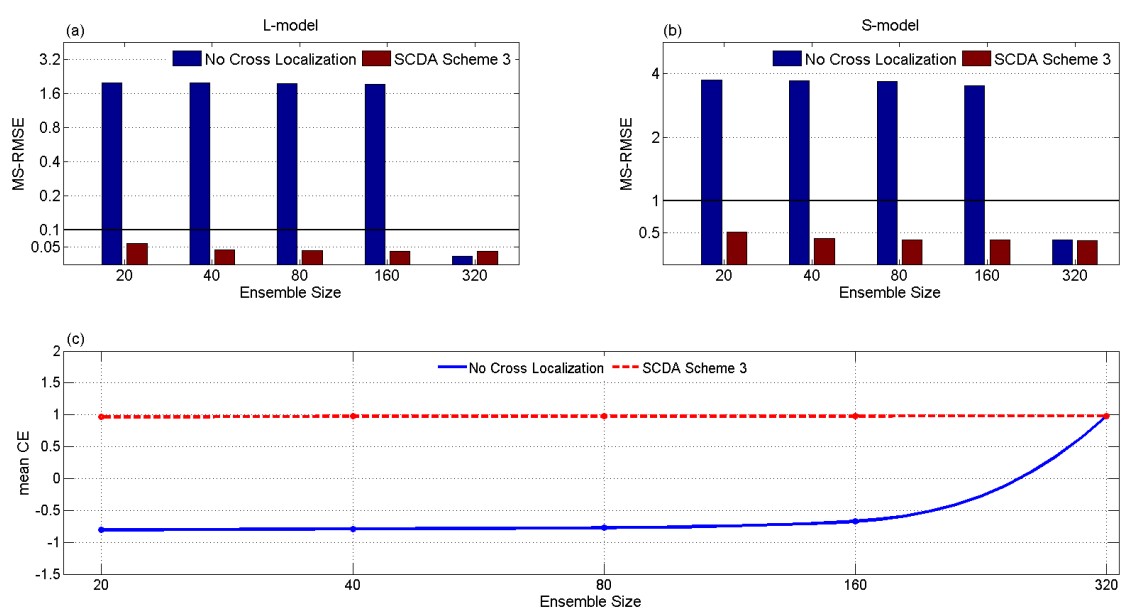

**Figure 9.** The MS-RMSE of L-model (a) and S-model (b) and the mean CE of the coupled system (c) against different ensemble sizes using no cross localization method and SCDA scheme 3. To display large MS-RMSE values, logarithmic coordinate is used beyond 0.1 and 1 in (a) and (b) respectively.