# Peer review of "On the localization in strongly coupled ensemble data assimilation using a two-scale Lorenz model"

_Nonlinear Processes in Geophysics, 2018_

## Referee Comment (RC1) · Anonymous Referee #1 · 14 Jan 2019

The paper discusses localization in strongly coupled data assimilation systems. The authors use a two-scale Lorenz model to develop their technique and prove their idea through twin-experiments. 1- In general, the paper is not well-written. The grammar and the structure of the sentences is quite weak. There are a lot of language mistakes that I can't even list here, for brevity. If this to be reviewed and submitted again, I strongly encourage editing the manuscript by a native speaker. 2- Furthermore, around 80% of the paper is based only on academic information that the community is aware of, with little to no novelty. The only new idea is found in Section 3.3 where the authors propose a cross-domain localization strategy for SCDA. However, even that seems to be model specific and may not be generalized for realistic atmospheric applications in

my opinion. 3- The assessment of the results is poor. I wasn't sure if the authors are reporting forecast or analysis errors. Also, the RMSE on its own is often not a very good metric to assess a new assimilation/localization strategy. I wanted to see how the authors choice of equations 10 and 11 affect the ensemble spread evolution. 4- Related to point #2, I suggest the authors test their method in a large model assimilating real data. This at least will improve the quality of the manuscript given the lack to any theoretical developments. 5- Sections 2.3 and 3.1 can be removed or summarized. The EAKF equations can be referenced and the same goes for Gaspari-Cohn function and localization in general. Based on the above evaluation, I recommend rejection of the paper!

---

## Referee Comment (RC2) · Anonymous Referee #2 · 24 Jan 2019

**General comments**

This manuscript considers (distance-based) localization in the ensemble adjustment Kalman filter (EAKF) for coupled data assimilation problems, with a particular focus on a localization scheme in a two-scale Lorenz model. Overall, the manuscript is clearly written and reasonably organized, and also contains some interesting findings and insights, e.g., results with respect to the combinations of different levels of coupling in assimilation, as reported in Table 1 and Figure 8. I am in support of the publication of the current work, after some minor issues are addressed.

[Figure]

**Specific comments**

1. Between lines 10 – 15, page 7. The authors mentioned that $P_{xx}$ is a diagonal-constant matrix. Does this mean that the off-diagonal elements in $P_{xx}$ are all zero? If so, it seems to me that very "strong" localization is applied in the EAKF. To see this, let's use notations similar to those in Eq. (7) of the manuscript, but here I dropped the index $n$ of ensemble members. Without loss of generality, and regardless of which type of EnKF is used, in general one would have the following update formula

$$\Delta x_m = \sum_{s=1}^{S} K_{m,s} \Delta y_s \,,\, m = 1, 2, , M; s = 1, 2, , S,$$

where $m$ and $s$ are the indices of model state variables and observation elements in the filter analysis scheme, respectively; $M$ and $S$ are the total numbers of model variables and observations; and $K_{m,s}$ is the element of the Kalman gain matrix on the $m$th row and $s$th column. In EAKF, when localization is conducted, it's equivalent to introducing some tapering coefficients $P_{m,s}$ to the update, such that

$$\Delta x_m = \sum_{s=1}^{S} P_{m,s} \, K_{m,s} \Delta y_s \,.$$

For the authors' specific problem in consideration, one has $M = S$. So "$P_{xx}$ is a diagonal-constant matrix" means that $P_{m,s} = 0$ if $m \neq s$, or in other words, the model variable $x_m$ is only updated using the observation at the same location as $x_m$. In reality, it may be possible that observations at nearby locations also contain certain information of $x_m$, so a "weaker" localization scheme may be useful. Although, I do see that, in this case, adding more observations in the update scheme may make localization much more complicated.

My suggestion here is thus to clarify the situation, and discuss the implication when $P_{xx}$ (likewise, $P_{zz}$) is chosen to be a diagonal matrix. (No action required for the side remark in the sequel) In general, it should be desirable to make the localization scheme more general and more flexible. For this purpose, the authors may wish to have a look at the idea behind the recently proposed correlation based adaptive localization.

**Technical corrections (minor issues)**

1. Line 22, page 6. In "...inversely proportional to the distances...", "proportional" does not sound accurate.

2. First line, page 7. In "...the prior ensemble member", consider adding "$n$-th" before "prior".

3. Page 7 mentions "correlation covariance" in a few places. I guess it should be "cross covariance" instead.

4. First line, page 8. In "could beyond...", add "be" before "beyond".

5. Line 21, page 9. Double check the notation $\rho_z Z$.

6. In Eq. (10), define the operator $\otimes$ before using it. It does not seem to be a standard tensor product (between two vectors).

7. Last line of Section 3, page 10. In "...L-variable has equal effect...", it seems "has" should be "have".

8. In the definition of CE (page 10), why $\overline{x^{true}}$ should be squared in the denominator.

9. Line 12, page 11. In "It is possible that the smaller MS-RMSE with SCDA in figure 7b is due to ...", it seems to me "figure 7b" should be "figure 6b" instead. Similarly, Line 20, in "whose results are shown in Figure 9d–f", maybe "Figure 9d–f" should be "Figure 8d–f".

10. Line 13, page 12. In "when N $\leq$ 320", should "$\leq$" should be "=" instead?

11. Line 24, page 12. In "...limited ensemble size", add "a" before "limited". Line 25, add "of" after "the presence".

12. Lines 5 – 6, page 13. Replace "factors" by "factor", and change "a update" to "an update".
* * *

---

## Author Comment (AC1) · 19 Mar 2019

According to the comments, we have a major revision of this manuscript, and we also hired a professional service to edit the English. We have point-to-point responses of the reviewers' comments as follow:

*The paper discusses localization in strongly coupled data assimilation systems. The authors use a two-scale Lorenz model to develop their technique and prove their idea through twin-experiments. 1- In general, the paper is not well-written. The grammar and the structure of the sentences is quite weak. There are a lot of language mistakes that I can't even list here, for brevity. If this to be reviewed and submitted again, I*

*strongly encourage editing the manuscript by a native speaker*

**RESPONSES:** We are sorry that the reviewer was not happy with our English. As non-native speakers, we are aware of our weakness. Thus, we have hired a professional service, American Journal Experts (AJE), to edit the English of this manuscript prior to submission, including the grammar, punctuation and spelling. We have attached the certificate at the end of this document. Moreover, we have submitted this reviewer comment to AJE, and they have performed an additional revision to ensure that the English of this manuscript is of high quality. We hope that this reviewer now finds the English in the revised version to be satisfactory.

*2- Furthermore, around 80 % of the paper is based only on academic information that the community is aware of, with little to no novelty. The only new idea is found in Section 3.3 where the authors propose a cross-domain localization strategy for SCDA. However, even that seems to be model specific and may not be generalized for realistic atmospheric applications in my opinion.*

**RESPONSES:** Usually, a new idea or method is developed based on previous work and existing methods. In this manuscript, we developed a new localization method for a strongly nonlinear coupling system using EAKF. It is typical to present an introduction to EAKF, an introduction to the model used, and some discussion of the possible concerns and problems for the current localization methods, which, according to this reviewer, may be considered academic information that the community is already aware of. However, we think this overview is necessary to not only provide a friendly presentation to the readers but also maintain scientific rigor. This approach is common in many scientific papers.

We did not count the frequency of these components in this manuscript, but we believe 80 % is slightly exaggerated. However, we think it is not important. This manuscript developed a new method for strongly coupled ensemble data assimilation, as the reviewer described. The use of SCDA systems is very new and highly challenging. New

ideas and methods used for the development of SCDA systems are difficult to imple-ment; this difficulty may be one of the main reasons why few researchers are working on data assimilation, especially SCDA, compared with the many researchers working on data analysis. We should encourage young scientists to work in such a challenging field. More importantly, we think the new method is original and novel, thereby making this manuscript sufficiently innovative and deserving of publication.

We agree with this reviewer that the developed method and its performance may be model-dependent. However, this is not a reason to reject the publication of this work since the model-dependence occurs not only for this work but also for many other pub-lished works. This is especially true in the development of data assimilation methods, which always use a simple Lorenz model to test and develop a new method at the first step. We can find many assimilation papers published in reputable journals that only use a Lorenz model to develop new ideas or methods. Thus, the Lorenz model is also called a "test bed" in the field of data assimilation method development, and its use has been widely published in the literature. Therefore, it is unfair if this work is rejected due to our use of a Lorenz model.

*3- The assessment of the results is poor. I wasn't sure if the authors are reporting forecast or analysis errors. Also, the RMSE on its own is often not a very good metric to assess a new assimilation/localization strategy. I wanted to see how the authors choice of equations 10 and 11 affect the ensemble spread evolution.*

**RESPONSES:** Thank you for this comment. Sorry for the confusion. We present both the forecast and analysis errors in the manuscript, as described in Section 2.2 (Page 4, line 28), which this reviewer may not have noticed.

For the revised manuscript, we reran the experiments with some new feature of EAKF and assessed the results using different metrics, as listed below.

1. We use the adaptive inflation scheme developed by Anderson (2009) in the EAKF.

The adaptive inflation method uses different inflation parameters for different state elements, and the parameters are updated according to the observation and localization factor as a function of time. By using this method in the experiment, we can exclude the influence of the inflation on the results. The comparison result can be attributed to the localization method, which reinforces the conclusions.

2. To assess the performance of the CDA methods on each model, we still use the mean of the scaled RMSE (MS-RMSE) and the "coefficient of efficiency" (CE), as described in the submission. In addition, to better assess the result, as this reviewer mentioned, we will also compute the ensemble spreads in each of the forecast and analysis steps of the experiment. The root-mean-squared spread is scaled by the long-term STD, and the mean of the scaled RMSS (MS-RMSS) is used to further evaluate the result of each CDA method, as shown in Figures 6 - 9 in the revised manuscript. The spread is a good measure of the uncertainty, and we think it will emphasize our conclusions.

Thank you for the suggestion.

*4- Related to point 2, I suggest the authors test their method in a large model assimilating real data. This at least will improve the quality of the manuscript given the lack to any theoretical developments.*

**RESPONSES:** Thank you for this suggestion. We will attempt to implement this method in a real model in the next step. However, it is beyond the scope of this manuscript, as argued above. Moreover, assimilating real data in a large coupled model is very time consuming and challenging. We also desire for the method to be published soon, which we believe will benefit and promote the study of the SCDA method.

*5- Sections 2.3 and 3.1 can be removed or summarized. The EAKF equations can be referenced and the same goes for Gaspari-Cohn function and localization in general.*

[Figure]

**RESPONSES:** Thank you for this comment. Section 2.3 briefly introduces the EAKF method as a convenience to readers who have little knowledge about EAKF and for the efficient discussion of the new method, which we believe to be reader friendly. Thus, we modified this section but still kept it in the revised version. In the revised manuscript, we also included the adaptive inflation method used in the experiments. As suggested, we only provided a reference (Anderson 2009) and did not discuss it.

This work addresses localization. Thus, we think the discussion of the localization in a general framework is essential, as it provides the necessary background for the development of the new method and further discussion.

Additionally, we think that the equation on the GC function should remain in Section 3.1. It is frequently referred in Section 3.2 and Section 3.3, which both discuss some important issues about the GC function and its applications.

Please also note the supplement to this comment:
https://www.nonlin-processes-geophys-discuss.net/npg-2018-50/npg-2018-50-AC1-supplement.pdf
* * *
[Figure]

**Fig. 1.** figure6: The MS-RMSE of L-model (a) and S-model (c), the MS-RMSS of L-model (b) and S-model (d), and the mean CE of the coupled system (e) against different ensemble sizes with WCDA and SCDA.

[Figure]

**Fig. 2.** figure7: The absolute analysis errors and ensemble spreads of the L-model using SCDA and WCDA (left) and of the S-model using SCDA and WCDA (right).

[Figure]

**Fig. 3.** figure8: The MS-RMSE of the L-model (a) and S-model (c), the MS-RMSS of the L-model (b) and S-model (d), and the mean CE of the coupled system (e) against different ensemble sizes and different CDA

[Figure]

**Fig. 4.** figure9: The MS-RMSE of the L-model (a) and S-model (c), the MS-RMSS of the L-model (b) and S-model (d), and the mean CE of the coupled system (e) with different SCDA formula

**Supplement:**

AJE AMERICAN JOURNAL EXPERTS

**EDITORIAL CERTIFICATE**

This document certifies that the manuscript listed below was edited for proper English language, grammar, punctuation, spelling, and overall style by one or more of the highly qualified native English speaking editors at American Journal Experts.

**Manuscript title:**

On the localization in ensemble coupled data assimilation using a multi-scale Lorenz model

**Authors:**

Zheqi Shen, Youmin Tang, Xiaojing Li, Yanqiu Gao, Qian Zhong, Junde Li

**Date Issued:**

February 21, 2019

**Certificate Verification Key:**

FD6B-FB5F-EFE3-864E-01C8

[Figure]

This certificate may be verified at www.aje.com/certificate. This document certifies that the manuscript listed above was edited for proper English language, grammar, punctuation, spelling, and overall style by one or more of the highly qualified native English speaking editors at American Journal Experts. Neither the research content nor the authors' intentions were altered in any way during the editing process. Documents receiving this certification should be English-ready for publication; however, the author has the ability to accept or reject our suggestions and changes. To verify the final AJE edited version, please visit our verification page. If you have any questions or concerns about this edited document, please contact American Journal Experts at support@aje.com.

American Journal Experts provides a range of editing, translation and manuscript services for researchers and publishers around the world. Our top-quality PhD editors are all native English speakers from America's top universities. Our editors come from nearly every research field and possess the highest qualifications to edit research manuscripts written by non-native English speakers. For more information about our company, services and partner discounts, please visit www.aje.com.

---

## Author Comment (AC2) · 19 Mar 2019

*Specific comments*

*Between lines 10 – 15, page 7. The authors mentioned that $P_{xx}$ is a diagonal-constant matrix. Does this mean that the off-diagonal elements in $P_{xx}$ are all zero? If so, it seems to me that very "strong" localization is applied in the EAKF. To see this, let's use notations similar to those in Eq. (7) of the manuscript, but here I dropped the index n of ensemble members. Without loss of generality, and regardless of which type of EnKF*

[Figure]

*is used, in general one would have the following update formula*

$$\Delta x_m = \sum_{s=1}^{S} P_{m,s} K_{m,s} \Delta y_s$$

*For the authors' specific problem in consideration, one has $M = S$. So "$P_{xx}$ is a diagonal-constant matrix" means that $P_{m,s} = 0$ if $m = s$, or in other words, the model variable $x_m$ is only updated using the observation at the same location as $x_m$. In reality, it may be possible that observations at nearby locations also contain certain information of $x_m$, so a "weaker" localization scheme may be useful. Although, I do see that, in this case, adding more observations in the update scheme may make localization much more complicated. My suggestion here is thus to clarify the situation, and discuss the implication when $P_{xx}$ (likewise, $P_{zz}$) is chosen to be a diagonal matrix. (No action required for the side remark in the sequel) In general, it should be desirable to make the localization scheme more general and more flexible. For this purpose, the authors may wish to have a look at the idea behind the recently proposed correlation based adaptive localization.*

**RESPONSES:** We apologize for using the term "diagonal-constant matrix". In the revised version, we indicate that $P_{xx}$ is a $K * K$ Toeplitz matrix, which means that each diagonal of $P_{xx}$ has the same value, e.g., the main diagonal has the value $\rho(0, c)$ and the k-th diagonal has $\rho(k, c)$. Therefore, the off-diagonal elements are non-zero unless the distance exceeds a cut-off radius of $2 * c$. Again, we apologize for the misleading statements, and thank you very much for the suggestions.

*Technical corrections (minor issues)*

*1. Line 22, page 6. In "...inversely proportional to the distances...", "proportional" does not sound accurate.*

**RESPONSES:** Thank you for pointing this out; we changed the statement to "The value of the localization function decreases when the location of the state element moves

away from the observation site."

*2. First line, page 7. In "...the prior ensemble member", consider adding "n-th" before "prior".*

**RESPONSES:** This has been added. Thank you for the suggestion.

*3. Page 7 mentions "correlation covariance" in a few places. I guess it should be "cross covariance" instead.*

**RESPONSES:** We have changed this term to "covariance matrix".

*4. First line, page 8. In "could beyond...", add "be" before "beyond".*

**RESPONSES:** This has been added. Thank you for the suggestion.

*5. Line 21, page 9. Double check the notation $\rho_z z$*

**RESPONSES:** This is a typo. Thank you for pointing out this error. It has been corrected.

6. In Eq. (10), define the operator before using it. It does not seem to be a standard tensor product (between two vectors).

**RESPONSES:** We changed the notations in Eq. (10) and (11) and defined the Kronecker product in the first line of page 10.

*7. Last line of Section 3, page 10. In "...L-variable has equal effect...", it seems "has" should be "have".*

**RESPONSES:** Thank you for pointing out this error. It has been corrected.

*8. In the definition of CE (page 10), why xtrue should be squared in the denominator.*

**RESPONSES:** This is a typo. Thank you for pointing out this error. It has been corrected.

*Line 12, page 11. In "It is possible that the smaller MS-RMSE with SCDA in figure 7b*

*is due to ...", it seems to me "figure 7b" should be "figure 6b" instead. Similarly, Line 20, in "whose results are shown in Figure 9d–f", maybe "Figure 9d–f" should be "Figure 8d–f".*

**RESPONSES:** Thank you for pointing out this error. It has been corrected. Additionally, we have rerun the experiments according to the suggestion of another reviewer, and some of the figures are reproduced.

*Line 13, page 12. In "when $N \leq 320$", should "$\leq$" should be "=" instead?*

**RESPONSES:** In the revised manuscript, we compare the CDA scheme 3 with an alternative localization method to show the impact of Eq. (10) on the SCDA of the S-observations. Therefore, the whole paragraph has been rewritten.

*9. Line 24, page 12. In "...limited ensemble size", add "a" before "limited". Line 25, add "of" after "the presence".*

**RESPONSES:** This has been added. Thank you for the suggestion.

*10. Lines 5 – 6, page 13. Replace "factors" by "factor", and change "a update" to "an update".*

**RESPONSES:** We have changed this phrasing. Thank you for the suggestion.

We have hired native English-speaking editors to improve the English and hope that this strategy can eliminate those errors. Thank you for your help.